# Positive Allosteric Modulators of SERCA Pump Restore Dendritic Spines and Rescue Long-Term Potentiation Defects in Alzheimer’s Disease Mouse Model

**DOI:** 10.3390/ijms241813973

**Published:** 2023-09-12

**Authors:** Anastasiya Rakovskaya, Alexander Erofeev, Egor Vinokurov, Ekaterina Pchitskaya, Russell Dahl, Ilya Bezprozvanny

**Affiliations:** 1Laboratory of Molecular Neurodegeneration, Peter the Great St. Petersburg Polytechnical University, St. Petersburg 195251, Russia; rakovskaya_av@spbstu.ru (A.R.); alexandr.erofeew@spbstu.ru (A.E.); vinokurov.ek@edu.spbstu.ru (E.V.); pchitskaya_ei@spbstu.ru (E.P.); 2Neurodon Corporation, 9800 Connecticut Drive, Crown Point, IN 46307, USA; rdahl@neurodon.net; 3Department of Physiology, UT Southwestern Medical Center at Dallas, Dallas, TX 75390, USA

**Keywords:** SERCA, positive allosteric modulators, calcium, Alzheimer’s disease, beta-amyloid, dendritic spines, calcium imaging, long-term potentiation

## Abstract

Alzheimer’s disease (AD) is a neurodegenerative disorder that affects memory formation and storage processes. Dysregulated neuronal calcium (Ca^2+^) has been identified as one of the key pathogenic events in AD, and it has been suggested that pharmacological agents that stabilize Ca^2+^ neuronal signaling can act as disease-modifying agents in AD. In previous studies, we demonstrated that positive allosteric regulators (PAMs) of the sarco/endoplasmic reticulum Ca^2+^ ATPase (SERCA) pump might act as such Ca^2+^-stabilizing agents and exhibit neuroprotective properties. In the present study, we evaluated effects of a set of novel SERCA PAM agents on the rate of Ca^2+^ extraction from the cytoplasm of the HEK293T cell line, on morphometric parameters of dendritic spines of primary hippocampal neurons in normal conditions and in conditions of amyloid toxicity, and on long-term potentiation in slices derived from 5xFAD transgenic mice modeling AD. Several SERCA PAM compounds demonstrated neuroprotective properties, and the compound NDC-9009 showed the best results. The findings in this study support the hypothesis that the SERCA pump is a potential therapeutic target for AD treatment and that NDC-9009 is a promising lead molecule to be used in the development of disease-modifying agents for AD.

## 1. Introduction

Alzheimer’s disease (AD) is a neurodegenerative disorder that affects memory formation and storage processes. Several hypotheses about the causes of AD have been proposed, but the so-called “amyloid cascade hypothesis” is a dominant model of AD pathogenesis. It states that increased production of the amyloidogenic Aβ42 peptide (or an increase in the Aβ42:Aβ40 ratio) is driving AD, causing a decreased number of synapses and neuronal death [1]. Therefore, great efforts have been made to develop agents that can reduce Aβ production or eliminate Aβ from the brain. Recently, the FDA approved Aduhelm and Leqembi (monoclonal antibodies against Aβ developed by Biogen and Eisai) for AD. However, these approvals were based primarily on amyloid clearance endpoints, and clinical benefits of these antibodies appear to be very limited despite potential serious side effects reported for both [2,3,4,5]. Therefore, a substantial unmet need exists for the development of safer therapies for Alzheimer’s disease (AD) that can potentially offer clinical benefits to patients. It is noteworthy that the immune system interacts with synapses as well as immune cells and mediators participating in synapse elimination during development and contributing to synaptic plasticity during adulthood [6,7]. Therefore, in the context of drug development, it is imperative to consider their impact on the immune system.

An alternative approach to developing AD treatments is based on the “calcium hypothesis” of AD [8,9]. This hypothesis proposes that dysregulation in cellular calcium homeostasis is the main driving force of neurodegeneration in AD [8,9]. The pharmacological normalization of calcium signaling in neurons is a promising approach for the development of therapeutic agents, and its potential efficacy and applicability is demonstrated through the example of memantine [10]—a noncompetitive NMDA glutamate receptor antagonist. Memantine is also a dopamine agonist and increases dopaminergic transmission. In cognitive terms, memantine primarily increases attention and episodic memory. Memantine offers no benefit in mild AD, but it improves symptoms in patients with moderate to severe AD and exhibits a favorable safety and tolerability profile [11]. One potentially interesting and relatively unexplored Ca^2+^-related target is the sarco/endoplasmic reticulum calcium ATPase (SERCA) Ca^2+^ pump. Compounds from a new class of substances—positive allosteric modulators (PAM)—do not affect the basal activity of SERCA but are capable of enhancing the extraction of Ca^2+^ from the cytoplasm into the endoplasmic reticulum, with significant increases in calcium concentration. Such an effect will avoid undesirable effects on normal calcium-sensitive processes, thereby minimizing possible side effects while preventing toxic increases in cytoplasmic calcium concentrations. In our recent study, it was demonstrated that a novel SERCA PAM agent (compound NDC-1173) improved the performance in behavioral studies and offered a robust benefit in reducing ER stress in the APP/PS1 transgenic AD mouse model [12]. Encouraging results have also been obtained for this class of compounds in earlier studies with preclinical models of AD [13].

In this research, we aimed to study the activity of a new class of SERCA PAMs in vitro and their potential protective and therapeutic effects on in vitro and in vivo AD models. We describe the identification and evaluation of several novel positive allosteric modulators of the SERCA pump: NDC-9009, NDC-9033, NDC-9136, and NDC-9342. We demonstrate that these SERCA PAMs facilitate Ca^2+^ clearance from the cytoplasm during its pathological increase, protect hippocampal dendritic spines from amyloid toxicity-induced degenerative changes in vitro, and efficiently recover synaptic plasticity in brain slices from 6-month-old 5xFAD mice, an aggressive Aβ model of AD.

## 2. Results

### 2.1. Identification and Functional Analysis of Novel SERCA PAMs

Our previous work utilizing a high-throughput fluorescence resonance energy transfer (FRET) assay consisting of labeled SERCA and phospholamban enabled the identification of an aminoquinoline class of compounds that perturb SERCA conformation [14]. This finding is significant because of SERCA’s reputation as an intractable target and also because the only previously reported small-molecule SERCA activators were based on istaroxime, a steroid-based molecule with polypharmacology (specifically Na^+^,K^+^-ATPase inhibition), a narrow therapeutic index, and reported toxicity [15,16]. Identified compounds acted as either inhibitors or activators of SERCA, and a standard ATPase assay was employed to identify only activators. Medicinal chemistry optimization of this series identified the four SERCA PAMs: NDC-9009, NDC-9033, NDC-9136, and NDC-9342. The structures of these compounds and their basic biochemical properties are summarized in Table 1.

### 2.2. SERCA PAMs Enhance the Rate of Calcium Extraction from the Cytosol in HEK293T Cell Line

To analyze the effect of SERCA PAMs on the rate of calcium extraction from the cytosol, HEK293T cells were transfected with the protein calcium-sensor GCaMP using polyethyleneamine. Cells were placed in ADMEM with 2 mM of Ca^2+^, and then one of four different PAMs (NDC-9009, NDC-9033, NDC-9136, or NDC-9342) was added at a final concentration of 0.1 μM along with the ionomycin (Io) ionophore at a final concentration of 1 μM (Figure 1A). Analysis of the results showed that NDC-9009, NDC-9136, and NDC-9342 inhibited the increase in calcium levels caused by the addition of ionomycin: the peak value of the GCaMP signal for the control group was 7.2 ± 0.6; with the addition of NDC-9009, it was 4.78 ± 0.3 (***: *p* < 0.001); for NDC-9136, it was 4.9 ± 0.5 (**: *p* < 0.01); and for NDC-9342, it was 5.9 ± 0.4 (*: *p* < 0.05,) (Figure 1B) (Table 2). The rate of increase in cytosolic calcium was estimated by calculating the tangent of the slope (tanα). PAMs caused a tanα decrease: for NDC-9009, to 0.021 ± 0.002 (***: *p* < 0.001); for NDC-9136, to 0.016 ± 0.002 (***: *p* < 0.001); and for NDC-9342, to 0.026 ± 0.003 (***: *p* < 0.001) (Figure 1C) (Table 2).

It was confirmed that NDC-9009, NDC-9136, and NDC-9342 reduced the calcium concentration with its pathological increase. The rate of calcium extraction during SERCA activation was also evaluated, and the best results were also demonstrated by these three compounds. Next, it was necessary to evaluate the neuroprotective potential of SERCA PAMs in primary hippocampal neurons in conditions of amyloid toxicity, modeling Alzheimer’s disease.

### 2.3. Effects of SERCA PAMs on the Morphology of Dendritic Spines of Primary Hippocampal Neurons in Normal Conditions and in Conditions of Amyloid Toxicity

To assess the neuroprotective potential of positive allosteric modulators (PAMs), primary hippocampal neurons were visualized via calcium phosphate transfection with the GFP plasmid on day 7 of in vitro culture (DIV). Seventy-two hours before fixation, cells were treated with Aβ42 (final concentration of 0.1 μM) oligomers to model amyloid synaptotoxicity conditions in vitro. The cells were incubated with 0.1 μM of one of the four PAMs (NDC-9009, NDC-9033, NDC-9136, or NDC-9342) or an equal volume of DMSO for 24 h. At DIV 16-17, cultures were fixed, and an analysis of the dendritic spine morphology was performed by using confocal microscopy (2048 × 2048 pixels with a resolution of 0.032 µm/pixel). Image processing was performed using the SpineJ plugin in ImageJ software. The micrographs in Figure 2C illustrate changes in the morphology of the dendritic spines of the hippocampal neurons after addition of PAM in normal conditions and in conditions of amyloid toxicity, simulating AD in vitro. A decrease in the area of the head of the dendritic spines was demonstrated for NDC-9033 (0.511 ± 0.026 µm^2^) (*: *p* < 0.05) and NDC-9342 (0.430 ± 0.019 µm^2^) (***: *p* < 0.001) (Figure 2A), while increases were shown in the neck length for the other two PAMs: NDC-9009 (0.696 ± 0.044 µm) (***: *p* < 0.001) and NDC-9136 (0.654 ± 0.049 µm) (*: *p* < 0.05) (Figure 2B). However, the ratio of neck length to spine length increased only when exposed to NDC-9009 (36 ± 1%) (***: *p* < 0.001) (Figure 2E), and no differences in density were observed compared to the control (Figure 2D). The spines’ head area decreased to 0.481 ± 00.019 µm^2^ (*: *p* < 0.05) in conditions of amyloid toxicity, consistent with the data obtained and characteristic of the in vitro AD model (Figure 2F). NDC-9009, NDC-9136, and NDC-9342 demonstrated a neuroprotective effect in the condition of amyloid toxicity by increasing the head area of the dendritic spines to 0.576 ± 0.041 µm^2^ (#: *p* < 0.05), 0.560 ± 0.028 µm^2^ (#: *p* < 0.05), and 0.563 ± 0.031 µm^2^ (#: *p* < 0.05) (Figure 2F) (Table 2), respectively. The length of the neck of the dendritic spines increased in conditions of amyloid toxicity from 0.488 ± 0.029 μm to 0.757 ± 0.057 μm (***: *p* < 0.001); however, the addition of PAM did not show statistical differences in this criterion (Figure 2G). There were no statistical differences between the groups in the density of dendritic spines per 10 µm (Figure 2H). The ratio of neck length to spine length increased from 30 ± 1% to 39 ± 2% (***: *p* < 0.001) in conditions of amyloid toxicity, but the addition of PAM reduced this parameter: NDC-9033—up to 34 ± 2% (#: *p* < 0.05); NDC-9342—up to 35 ± 2% (#: *p* < 0.05) (Figure 2I) (Table 2).

Based on the results of the experiments summarized in Appendix A, it can be concluded that the NDC-9009 compound demonstrated the best neuroprotective effect and the highest efficiency, and thus it was chosen for further experiments to evaluate its neuroprotective potential in vivo.

### 2.4. Effects of NDC-9009 on Long-Term Potentiation in an AD Mouse Model

Hippocampal synaptic plasticity occurs in multiple phases involving short-term and long-term changes in the synapse [17]. Changes within 0–2 min after high-frequency stimulation (HFS) are thought to reflect post-tetanic potentiation associated with alterations in presynaptic neurotransmitter release properties [18]. Changes in the last 10 min after HFS stimulation are considered long-term potentiation [19]. We conducted experiments to assess post-tetanic potentiation (PTP) and long-term potentiation (LTP) in control and experimental groups of wild-type (WT) and Alzheimer’s mice (5xFAD) characterized by the production and aggregation of beta-amyloid protein (Figure 3; representative images of amyloid plaques in the brains of 6-month-old 5xFAD and WT mice are presented in Appendix A). Both the wild-type and AD experimental groups were administered intraperitoneal injections of NDC-9009 at a 10 mg/kg dose (DMSO + NDC9009:TWEEN 80:water) for one month starting at 5 months of age. The control group received a vehicle solution (DMSO:TWEEN 80:water).

Analysis of the PTP (2 min after high-frequency stimulation) revealed no difference in the fEPSP slope between the WT and 5xFAD control groups at 6 months old, with values of 299 ± 22 and 306 ± 24, respectively (Figure 4A). However, NDC-9009 administration significantly increased the PTP fEPSP slope of the 5xFAD mice in comparison to WT, with values of 340 ± 13 and 258 ± 20, respectively.

We next examined the LTP expression (last 10 min after high-frequency stimulation) in 6-month-old mice (Figure 4B). After HFS, the fEPSP slope increased over time in all groups. The increments in the slope of the fEPSP were slower in 5xAFD mice than in the wild-type mice in the control group and changed in the experimental group with NDC-9009. Normalized to baseline, the percentage of slope (LTP, last 10 min after HFS) values was lower in 5xFAD mice (145 ± 3) than in WT mice (169 ± 5) in the control group.

At the same time, significant differences (*: *p* < 0.05) were observed in the magnitude of long-term potentiation between the wild-type mice (153 ± 6) and 5xFAD mice (181 ± 7) in the experimental group. Furthermore, a significant difference (**: *p* < 0.01) in long-term potentiation was observed in both the control and experimental groups of 5xFAD mice. Specifically, the LTP value within the experimental group of 5xFAD mice exceeded that observed in the corresponding control group. In the case of the wild-type mice, no statistically differences in the long-term potentiation (LTP) value were discerned between the experimental and control groups of mice. 

Based on our findings, it can be inferred that the positive allosteric modulator of calcium ATPase SERCA NDC-9009 exhibits a neuroprotective effect. The intraperitoneal administration of NDC-9009 led to an increase in the magnitude of long-term potentiation in 6-month-old 5xFAD mice compared to the control group of mice.

## 3. Discussion

Alzheimer’s disease (AD) is a neurodegenerative disorder that affects memory formation and storage processes. Several hypotheses about the causes of AD have been proposed, but the so-called “amyloid cascade hypothesis” is a dominant model of AD pathogenesis. It states that increased production of amyloidogenic Aβ42 peptide is driving AD, causing a decreased number of synapses and neuronal death [1]. As an alternative viewpoint, the “calcium hypothesis” of AD has also been suggested [20]. This hypothesis assumes that dysregulation in cellular calcium homeostasis mechanisms is the main driving force of neurodegeneration in AD [8,9].

In addition to the adverse effect of amyloid β-protein, Ca^2+^ dysregulation in AD is caused by mutated presenilins [21]. Different cellular models expressing AD mutant presenilin demonstrate an overloading of the endoplasmic reticulum (ER) with Ca^2+^ and an excessive Ca^2+^ release through the InsP_3_R [22,23,24,25], altering parts of the neuronal calcium signaling machinery such as the store-operated Ca^2+^ influx [26] and induction of excessive ryanodine receptor (RyR) calcium release [25]. Most of the AD-causing mutations block the pore formed by presenilin, leading to ER calcium overload [22,23], which subsequently results in excessive calcium release through the RyR [25]. A reduction in the RyR calcium release was proposed as a therapeutic strategy for AD treatment. However, experiments with its inhibitor (dantrolene) have yielded contradictory results, possibly due to its lack of selectivity [27,28]. Recent findings suggest that the neuronal-store-operated calcium entry (SOCE) pathway plays a role in AD pathogenesis. The SOCE pathway is activated following depletion of ER Ca^2+^ levels and is reduced in AD. The most obvious reason for SOCE alteration is the reduced expression level of the ER Ca^2+^ sensor STIM2/STIM1 [29,30,31]. Pharmacological restoration of the SOCE pathway is one potential method for drug development in AD [29].

The primary focus of this article centers on the SERCA Ca^2+^ pump. SERCA’s well-established role is to preserve low cytosolic Ca^2+^ levels by pumping free Ca^2+^ ions into the ER lumen by utilizing ATP hydrolysis. It has been suggested that presenilins can directly interact with SERCA and physiologically regulate its activity and confer resistance to endoplasmic reticulum stress [32]. SERCA ensures proper Ca^2+^ handling in cells and may act as a therapeutic target for the disease associated with dysregulation of calcium ions [33]; therefore, development of its selective activators is under scientific investigation. Recently, a small transmembrane SERCA-binding protein known as the Dwarf Open Reading Frame (DWORF) was discovered as a direct activator of SERCA. This fundamental discovery about the physiological regulation of SERCA holds promise for the development of a new generation of its activators [34]. Also, a study conducted on a pyridone derivative was found to activate the SERCA2a isoform [35]. Furthermore, it was observed that the pyridone derivative stimulated the Ca^2+^-dependent ATPase activity of cardiac sarcoplasmic reticulum (SR) vesicles. This indicates that the derivative enhances the ability of SR to transport calcium ions, which is an essential process for proper cardiac function. This results in improved cardiac function at both the cellular and organ level. Therefore, this pyridone derivative is suggested as a promising candidate for therapeutic applications in heart failure [35]. Another study demonstrated that CDN1163, when activating the SERCA2b isoform in the liver, can reduce endoplasmic reticulum (ER) stress and improve mitochondrial efficiency. Additionally, metabolic parameters were improved, suggesting that CDN1163 or other SERCA activators have the potential to be pharmacological agents for treating diabetes and metabolic dysfunction [14,36]. Additionally, small-molecule activation of SERCA2 by the quinoline derivative CDN1163 [37] was supposed to be a potential pharmacotherapeutic target in Alzheimer’s and Parkinson’s diseases [38]. CDN1163 has been demonstrated to be effective in the APP/PS1 double transgenic mouse model of Alzheimer’s disease [13].

Previous studies confirmed that SERCA PAM CDN1163 can exert beneficial effects in APP/PS1 mice [13]. Our recent paper also concluded that another SERCA PAM—compound NDC-1173—could enhance memory in both object and spatial memory tasks in an AD mouse [12]. In the present study, we report the activity of another SERCA PAM molecule—compound NDC-9009. When compared to other molecules (NDC-9033, NDC-9136, and NDC-9342), this compound demonstrated the most consistent effects on Ca^2+^ dynamics (Figure 1) and in the spine rescue assay (Figure 2) in hippocampal neuronal cultures (Table 2). We demonstrated the in vitro calcium ATPase activity using SERCA PAMs on the HEK293T cell line. Additionally, we evaluated the neuroprotective potential of SERCA PAMs in conditions of amyloid toxicity, which is believed to be the primary driver of AD pathology. NDC-9009 also exhibited effectiveness in the LTP rescue assay using hippocampal slices from 5xFAD mice (Figure 3 and Figure 4). It is worth noting that we investigated alterations in long-term potentiation in 6-month-old 5xFAD male mice. We used exclusively male mice in our research and examined a single concentration of the compound. To gain a more comprehensive understanding of the compound under investigation, further experiments involving a range of concentrations and both male and female mice are warranted. Based on these findings, we concluded that NDC-9009 is the most promising SERCA PAM from this compound series for the development of potential AD therapeutic agents.

However, several limitations of this study should be noted. First, future studies are needed to analyze another system (heart or pancreas) after using SERCA PAMs in vivo to rule out potential side effects. Second, we focused our study on the 5xFAD mice model, which is a very aggressive model of AD and has its critics [39]. Testing NDC-9009 or other SERCA PAMs in “second generation” AD models such as APPKI mice [40] may provide further insights into their mechanism of action.

## 4. Materials and Methods

### 4.1. Synthesis of SERCA PAMs

NDC-9009. Into a stirred solution of 4-(dimethylamino) benzoic acid (2.50 g, 15.134 mmol) and DMF (50.00 μL) in DCM (30.0 mL), oxalyl chloride (9 mL, 2 M in DCM) was added dropwise at 0 °C under nitrogen. The resulting mixture was stirred for 1 h at room temperature and then concentrated under reduced pressure. The crude product was used in the next step directly without further purification. The residue was dissolved in DCM (30.0 mL) and added dropwise into a stirred solution of 2-methylquinolin-8-amine (1.30 g, 8.217 mmol) and TEA (3.33 g, 32.868 mmol) in DCM (30.0 mL) at 0 °C. The resulting mixture was stirred for 2 h at room temperature under a nitrogen atmosphere. LCMS showed that the SM was consumed and the product was formed. The mixture was washed with water, extracted with DCM, and dried over anhydrous Na_2_SO_4_. After filtration, the filtrate was concentrated under reduced pressure. The residue was purified via silica gel column chromatography eluting with EtOAc/DCM (0–5%) to afford the crude product. The residue was dissolved in DCM (50 mL) and washed with 1N of HCl (50 mL). The aqueous layer was adjusted to pH 8 with NaHCO_3_ and extracted with DCM (50 mL). The organic layers were dried over anhydrous Na_2_SO_4_ and concentrated under reduced pressure. The residue was recrystallized from PE: EtOAc = 10:1 (50 mL) to afford 1.46 g (58% yield) of the title compound as a light yellow solid. LC-MS: (ESI, *m*/*z*): [M + H]^+^ = 306. ^1^H NMR (400 MHz, DMSO-*d*_6_) *δ* 10.58 (s, 1H), 8.71–8.67 (m, 1H), 8.31 (d, *J* = 8.4 Hz, 1H), 7.88 (d, *J* = 9.2 Hz, 2H), 7.63–7.50 (m, 3H), 6.86 (d, *J* = 8.8 Hz, 2H), 3.03 (s, 6H), 2.78 (s, 3H).

NDC-9033. Into a stirred solution of 3-isopropoxybenzoic acid (1.80 g, 9.989 mmol) and DMF (50.00 μL, 0.684 mmol) in DCM (30.0 mL), oxalyl chloride (6 mL, 2 M in DCM) was added dropwise at 0 °C under nitrogen. The resulting mixture was stirred for 1 h at room temperature and then concentrated under reduced pressure. The crude product was used in the next step directly without further purification. The residue was dissolved in DCM (30.0 mL) and added dropwise into a stirred solution of 2-methylquinolin-8-amine (1.74 g, 10.998 mmol) and TEA (2.02 g, 19.978 mmol) in DCM (30.0 mL) at 0 °C. The resulting mixture was stirred for 2 h at room temperature under a nitrogen atmosphere. LCMS showed that the SM was consumed and the product was formed. The mixture was washed with water and dried over anhydrous Na_2_SO_4_. After filtration, the filtrate was concentrated under reduced pressure. The residue was purified via silica gel column chromatography eluting with EtOAc/DCM (0–5%) to afford the crude product. The crude product was further purified in a pre-packed C18 column (solvent gradient: 0–50% ACN in water (10 mmol/L NH_4_HCO_3_)) to afford 1.92 g (59% yield) of the title compound as a white solid. LC-MS: (ESI, *m*/*z*): [M + H]^+^ = 321. ^1^H NMR (400 MHz, DMSO-*d_6_*) *δ* 10.68 (s, 1H), 8.70–8.67 (m, 1H), 8.33 (d, *J* = 8.4 Hz, 1H), 7.70–7.66 (m, 1H), 7.59–7.52 (m, 5H), 7.23–7.19 (m, 1H), 4.78–4.72 (m, 1H), 2.75 (s, 3H), 1.34 (d, *J* = 6.0 Hz, 6H).

NDC-9136. Into a stirred solution of 4-isopropoxybenzoic acid (1.80 g, 9.989 mmol) and DMF (50.0 μL) in DCM (30.0 mL), oxalyl chloride (6.0 mL, 2 M in DCM) was added dropwise at 0 °C under nitrogen. The resulting mixture was stirred for 1 h at room temperature and then concentrated under reduced pressure. The crude product was used in the next step directly without further purification. The residue was dissolved in DCM (30.0 mL) and added dropwise into a stirred solution of 2-methylquinolin-8-amine (1.70 g, 10.746 mmol) and TEA (2.17 g, 21.445 mmol) in DCM (30.0 mL) at 0 °C. The resulting mixture was stirred for 2 h at room temperature under a nitrogen atmosphere. LCMS showed that the SM was consumed and the product was formed. The resulting mixture was washed with water and dried over anhydrous Na_2_SO_4_. After filtration, the filtrate was concentrated under reduced pressure. The residue was purified via silica gel column chromatography eluting with EtOAc/DCM (0–5%) to afford the crude product. The crude product was dissolved in DCM (50 mL) and washed with 1N HCl (50 mL) and 10% NaHCO_3_ (50 mL). The organic layers were dried over anhydrous Na_2_SO_4_. After filtration, the filtrate was concentrated under reduced pressure to afford 1.8 g (52% yield) of the title compound as a white solid. LC-MS: (ESI, *m*/*z*): [M + H]^+^ = 321. ^1^H NMR (400 MHz, DMSO-*d_6_*) *δ* 10.63 (s, 1H), 8.71–8.67 (m, 1H), 8.33 (d, *J* = 8.4 Hz, 1H), 8.01–7.95 (m, 2H), 7.68–7.63 (m, 1H), 7.59–7.52 (m, 2H), 7.13 (d, *J* = 9.2 Hz, 2H), 4.82–4.72 (m, 1H), 2.78 (s, 3H), 1.32 (d, *J* = 6.0 Hz, 6H).

NDC-9342. To a stirred solution of 4,5-dibromothiophene-2-carboxylic acid (7.00 g, 24.5 mmol) in DCM (70 mL), DMF was added (47.2 mg, 0.646 mmol), and the mixture was allowed to cool down to 0 °C. Then, oxalic dichloride (6.30 g, 49.6 mmol) was added dropwise at 0 °C under a nitrogen atmosphere. The resulting mixture was stirred for 2 h at room temperature under a nitrogen atmosphere. The resulting mixture was concentrated under reduced pressure. The crude product was used in the next step directly without further purification. The residue was dissolved in DCM (30 mL) and added dropwise into a stirred solution of 2-methylquinolin-8-amine (4.40 g, 27.8 mmol) and TEA (7.10 g, 70.2 mmol) in DCM (40 mL) at 0 °C. The resulting mixture was stirred for 2 h at room temperature under a nitrogen atmosphere. The desired product could be detected via LC-MS. The mixture was washed with 100 mL of water, and the aqueous phase was extracted with DCM (80 mL). The combined organic layers were concentrated under reduced pressure. The residue was purified via silica gel column chromatography and eluted with PE/EA = 63/37 to afford 9.00 g of the product. Then, 30 mL of EA was added carefully into 4,5-dibromo-N-(2-methylquinolin-8-yl)thiophene-2-carboxamide (purity >95%) to form a light yellow slurry. The resulting slurry was stirred for 36 h at room temperature. After filtration, the solid was collected and dried overnight at 40 °C to afford 4,5-dibromo-N-(2-methylquinolin-8-yl)thiophene-2-carboxamide (6.55 g, 62%) as an off-white solid. LC/MS (ESI, *m*/*z*): [M + H]^+^ = 426.6. ^1^H NMR (300 MHz, CDCl_3_) δ 10.61 (s, 1H), 8.72 (dd, *J* = 6.2, 2.8 Hz, 1H), 8.10 (d, *J* = 8.4 Hz, 1H), 7.56–7.47 (m, 3H), 7.38–7.36 (m, 1H), 2.78 (s, 3H).

### 4.2. Biochemical Analysis of SERCA PAMs

SERCA Activity Assay. ATPase assays in HEK 293 lysates were carried out using the ATPase Assay Kit (Colorimetric; Cat. No. ab234055; Abcam, Cambridge, UK) as per the manufacturer’s instructions. Briefly, HEK 293 cells and either DMSO as a control or 2 µM of the test compound were incubated in the provided assay buffer containing ATPase substrate and developed at 25 °C for 20 min, and the A_650_ was measured using an Accuris SmartReader 96 Microplate Reader (Benchmark Scientific Inc., Sayreville, NJ, USA).

### 4.3. Animals

Albino inbred mice (FVB/NJ) were obtained from the Jackson Laboratory (Jackson Laboratory, Bar Harbor, ME, USA, strain #001800) and used as a source of brain tissue for experiments with primary hippocampal cultures. The 5xFAD mice (Jackson Laboratory, Bar Harbor, ME, USA, strain #034840-JAX) in a B6SJLF1 background were obtained from the Jackson Laboratory and used as a source of brain slices for LTP induction experiments. These mice were established and maintained in a vivarium with 4 to 5 mice per cage and a 12-hour light/dark cycle in the animal facility. Food and water were available ad libitum. All procedures adhered to the principles of the European Convention and the Declaration of Helsinki regarding the humane treatment of animals and were approved by the Bioethics Committee of Peter the Great St. Petersburg Polytechnic University in St. Petersburg, Russia (Ethical Permit No. 2-n-b from 25 January 2021).

### 4.4. Primary Hippocampal Cultures, Calcium Phosphate Transfection, and Immunohistochemistry

Primary hippocampal neuronal cultures of dissociated hippocampal cells were prepared from newborn FVB mice and maintained in culture as described previously [31]. Briefly, the hippocampus from postnatal day 0–1 was incubated with papain solution for 30 min at 37 °C (Worthington Biochemical Corp., Lakewood, NJ, USA, #3176) then dissociated with a solution of deoxyribonuclease I (5 mg/mL; Macherey Nagel GMBH, Germany, #R1542S). The neurons were placed in a 24-well plate on 12 mm glass cover glasses pre-coated with 1% poly-D-lysine (Sigma-Aldrich, St Louis, MO, USA, #p-7886). Cells were grown in 1 mL Neurobasal-A (Thermo Fisher Scientific, Waltham, MA, USA, #10888022) supplemented with 2% B27 (Thermo Fisher Scientific, Waltham, MA, USA, #17504044), 1% heat-inactivated fetal bovine serum (FBS, Thermo Fisher Scientific, Waltham, MA, USA, #10500064), and 0.5 mM L-glutamine (Thermo Fisher Scientific, Waltham, MA, USA, #25030024), and were maintained at 37 °C in a 5% CO_2_ incubator in a 24-well glass plate. Transfection was performed according to [41] with a calcium transfection kit purchased from Clontech (Takara Bio, Kusatsu, Japan, #631312) with GFP plasmid. Preparation of the oligomeric beta-amyloid was described in [31]. Cells were incubated with Aβ for 72 h. The cells were incubated with 0.1 μM of one of the four PAMs (NDC-9009, NDC-9033, NDC-9136, or NDC-9342) or an equal volume of DMSO for 24 h before fixation.

### 4.5. HEK293T Cultures and Transfection with Polyethyleneamine

HEK293T line cells with 50–70% of confluency were co-transfected with the GCaMP5.3 plasmid using a polyethylenimine reagent (Polysciences Inc., Warrington, PA, USA, #23966) in serum-free Opti-MEM medium (Thermo Fisher Scientific, Waltham, MA, USA, #11058-021). After 3 h of incubation in a CO_2_ incubator, the Opti-MEM was replaced with full DMEM (Thermo Fisher Scientific, Waltham, MA, USA, #41965-039; 10% FBS; 1% Pen Strep, Thermo Fisher Scientific, Waltham, MA, USA, #15140-122). HEK293T cells were grown to a confluence of 40% to 50% on round coverslips (12 mm diameter) in one well of a 24-well culture plate.

### 4.6. Intraperitoneal Administration

In order to evaluate the neuroprotective efficacy of a positive allosteric modulator on long-term potentiation (LTP), NDC9009 was administered intraperitoneally (10 mg/kg; DMSO + NDC9009:TWEEN80:water at a 10:10:80 ratio) with daily injections for 4 weeks (5 days every week) starting at 5 months [13]. Control mice were administered the vehicle, which consisted of DMSO:TWEEN80:water at a 10:10:80 ratio. The daily injections occurred for 4 weeks on weekdays only starting when the mice were 5 months of age.

### 4.7. Long-Term Potentiation Recording

For the long-term potentiation experiments, male wild-type mice and the 5xFAD mouse line were utilized. At 6 months of age, acute living hippocampal slices from injected mice were prepared, and field excitatory postsynaptic potentials (fEPSPs) were recorded to evaluate the magnitude of long-term potentiation. For this purpose, mice were anesthetized with isoflurane (1.5–2%) and perfused transcardially with ice-cold modified artificial cerebrospinal fluid containing (in mM) 25 D-glucose, 5 HEPES, 124 NaCl, 2.5 KCl, 3 Na-pyruvate, 1.25 NaH_2_PO_4_·2H_2_O, 24 NaHCO_3_, 0.5 CaCl_2_·2H_2_O, and 10 MgSO_4_·7H_2_O (pH 7.3–7.4). Brains were rapidly removed into ice-cold modified artificial cerebrospinal fluid. The 400 μm thick slices were cut on a vibratome (Leica Microsystems, Wetzlar, Germany) and immediately transferred to an incubation chamber. The slices were incubated to recover at 32 °C for 30 min and transferred to an incubation chamber artificial cerebrospinal fluid containing (in mM) 25 D-glucose, 5 HEPES, 124 NaCl, 2.5 KCl, 1.25 NaH_2_PO_4_·2H_2_O, 24 NaHCO_3_, 2.6 CaCl_2_·2H_2_O, and 1.3 MgSO_4_·7H_2_O (pH 7.3–7.4) for 60 min at room temperature. During the slice preparation and recording, artificial cerebrospinal fluid was continuously mixed with the gas, including 95% O_2_ and 5% CO_2_. Following incubation at room temperature, the hippocampus slice was transferred to the recording chamber and allowed to equilibrate for an additional 20 min prior to the recording of fEPSPs.

The stimulating bipolar electrode was prepared by double-twisting two platinum-iridium wires with a diameter of 50 microns (A-M Systems Inc., Carlsborg, WA, USA). The CA1 region located at the boundary between the CA1 and CA2 regions of the hippocampus was chosen as the stimulation site; the recording site was determined by the occurrence of an fEPSP response in the CA1 area. The fEPSPs were evoked by a Model 2100 isolated pulse stimulator (A-M Systems Inc., Carlsborg, WA, USA); the stimulation current (from 10 µA in steps of 5–10 µA) was adjusted accordingly until the occurrence of a population spike. The evoked fEPSPs were amplified with a MultiClamp 700B (Molecular Devices, San Jose, CA, USA) and digitized with a Digidata 1440A (Molecular Devices, San Jose, CA, USA) at a 20 kHz sampling rate. Baseline responses were stabilized (fEPSP slope deviation of less than 10%) and then recorded for 20 min at 0.05 Hz of stimulation. Long-term potentiation was then induced by high-frequency stimulation (HFS), which consisted of two bursts with an interval of 20 s; each burst had a duration of 1s and a stimulation frequency of 100 Hz. After HFS, the stimulus was repeatedly delivered once every 20 s for 60 min to observe any changes in the LTP magnitude. The LTP magnitude was defined as the ratio between the average fEPSP slope after HFS during the last 10 min and the average slope of the baseline fEPSPs. Post-tetanic potentiation (PTP) was defined as the ratio between the average fEPSP slope after HFS during the first 2 min and the average slope of the baseline fEPSPs. Traces were obtained and analyzed using the pClamp 10.7 software program (Molecular Devices, San Jose, CA, USA).

### 4.8. Analysis of Dendritic Spine Morphology in Primary Hippocampal Cultures

For assessment of the dendritic spine morphology, a Z-stack of the optical section was captured with a confocal microscope (Leica TCS SP8). For dendritic analysis, 2048 × 2048-pixel images with a 0.032 μm/pixel resolution were captured with Z interval of 0.2 μm using a 10× objective lens (NA = 0.85, UPlanSApo; Olympus, Tokyo, Japan). The images were pre-processed with ImageJ’s built-in “Median” filter to remove noise. Quantitative analysis of the dendritic spines, including measurements of the dendritic spine head area, neck length, and neck length/dendritic spine length ratio, was performed using SpineJ v1.0 software [42].

### 4.9. Calcium Imaging

Cells of the HEK293T line were transfected with a GCaMP5.3 expression plasmid using polyethyleneamine, as described earlier. Live fluorescent images were collected every 0.5 s using an Olympus IX73 fluorescent microscope with a 10× objective (UPlanFL N, Olympus, Tokyo, Japan) and a ZYLA 4.2 sCMOS camera (Andor Technology Ltd, Belfast, UK). Cells were incubated in 2 M Ca^2+^ ADMEM solution (110 mM NaCl, 5.3 mM KCl, 25 mM D-glucose, and 10 mM HEPES; pH  =  7.4) for 10 min. After recording basal fluorescent signals in the 2 M Ca^2+^ ADMEM solution for 30 s, release of ER Ca^2+^ was induced via puff application of 0.2 µL of ionomycin and 1 µL of one of the four PAMs (NDC-9009, NDC-9033, NDC-9136, or NDC-9342). Analysis of the data was performed using ImageJ software. The ROI used in the image analysis was chosen to correspond to spines, and the signal was normalized to the baseline.

### 4.10. Statistical Analyses

Statistical data analysis was performed using R studio software and GraphPad Prism software. To determine the distribution, the Shapiro–Wilk test or Lillifors test was calculated. If the distribution was not normal, the Kruskal–Wallis test was used. For multiple pairwise comparisons, Dunn’s nonparametric test or the Conover–Iman test were used. The one-way ANOVA test was employed to analyze the data assuming a normal distribution. Subsequently, the Tukey test was applied to facilitate multiple pairwise comparisons. Statistical tests are indicated in the figure legends. The data are presented as the mean ± S.E.M.

## Figures and Tables

**Figure 1 ijms-24-13973-f001:**
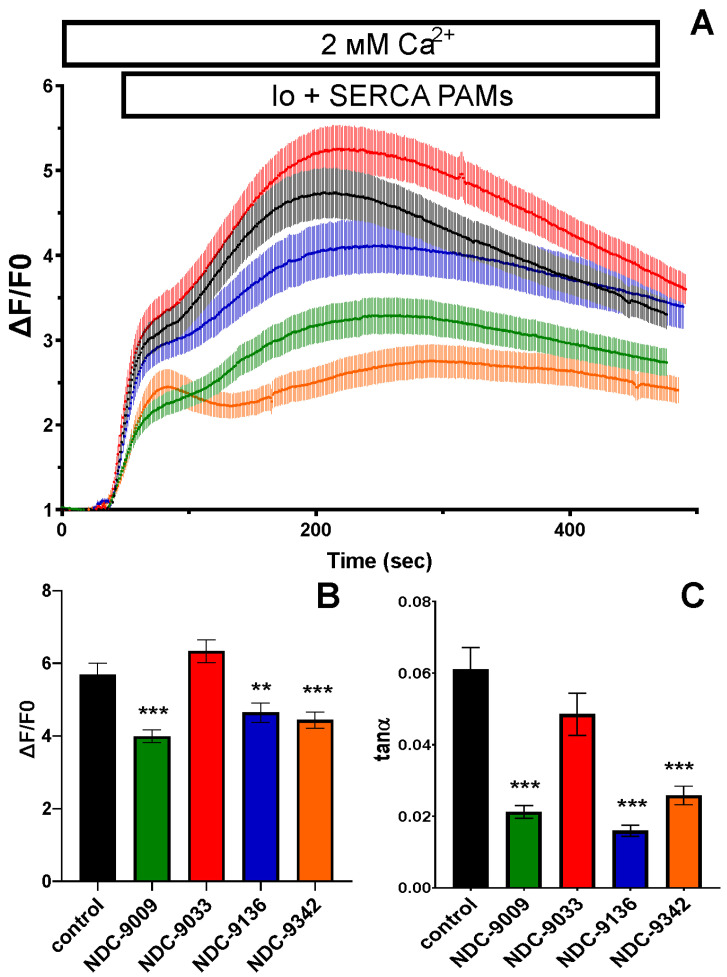
SERCA PAMs increased the rate of calcium extraction from the cytoplasm of the HEK293T cell line. (**A**) Calcium signal dF/F0 after addition of 1 μM of ionomycin (Io) in the control group and with the addition of PAMs. (**B**) Peak value of dF/F0 and (**C**) curve slope (tanα) for the control group and with the addition of PAMs. Data are presented as the mean ± SEM (*n*  ≥  15 cells from 3 batches of cultures); Dunn’s test was used for multiple comparison (**: *p* < 0.01,***: *p* < 0.001 compared to control).

**Figure 2 ijms-24-13973-f002:**
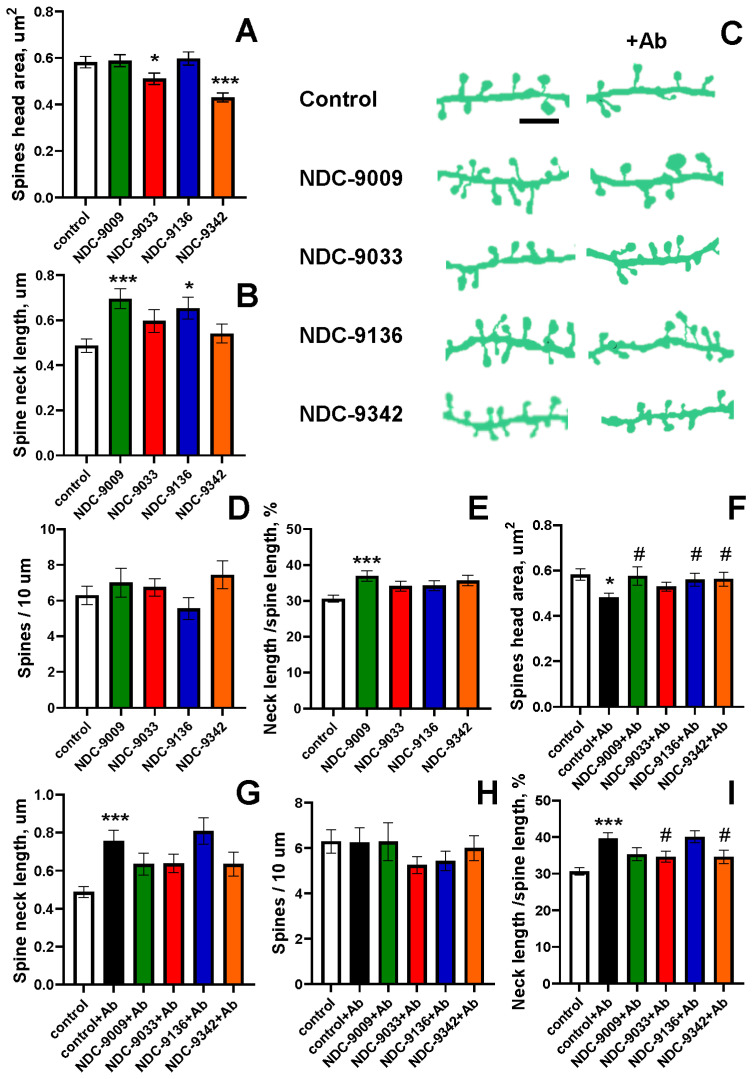
Positive allosteric modulators of calcium ATPase SERCA have a neuroprotective effect on hippocampal neurons in conditions of low amyloid toxicity. (**A**,**B**,**F**,**G**) The spines’ head area and spine length neck for each group of cells shown in panel C in the normal conditions (**A**,**B**) and in conditions of amyloid toxicity (Ab) (**F**,**G**) are presented as the mean  ±  SEM (*n*  ≥  10 neurons from 1 batch of cultures). (**C**) Binarized images of dendrites of wild-type (WT) hippocampal neurons transfected with GFP plasmid at DIV 7; one of four PAMs were added 24 h before fixation, and the culture was fixed at DIV 16-17. Scale bar corresponds to 5 μm. (**D**,**E**,**H**,**I**) Number of spines per 10 μm dendrite length and the ratio of the length of the neck to the length of the dendrite for each group of cells shown in panel C in normal conditions (**D**,**E**) and in conditions of amyloid toxicity (Ab) (**H**,**I**) are presented as the mean  ±  SEM (neurons  ≥  10, spines  ≥  70 from 1 batch of cultures). Statistical analysis was performed using a Conover–Iman test (*: *p* < 0.05, ***: *p* < 0.001 compared to control; #: *p* < 0.05 compared to control + Ab).

**Figure 3 ijms-24-13973-f003:**
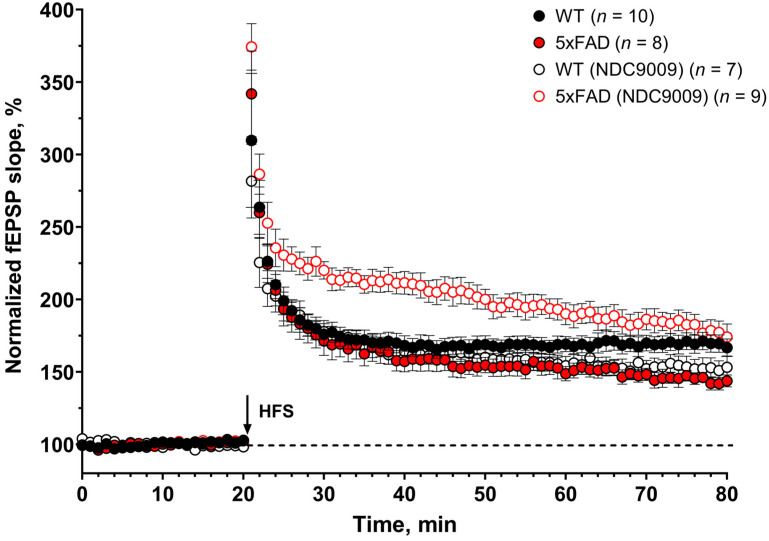
Positive allosteric modulators of SERCA pump (NDC-9009) restores LTP impairment in 6-month-old 5xFAD mice. The figure shows summary plots of the normalized fEPSP slope (%) in the recording time. The fEPSPs were induced by HFS (black arrow). After HFS, the fEPSP slope increased over time in all groups. The increments in the slope of the fEPSP were slower in 5xAFD mice than the wild-type (WT) mice in the control group and changed in the experimental group with NDC-9009. Data are presented as the mean ± SEM.

**Figure 4 ijms-24-13973-f004:**
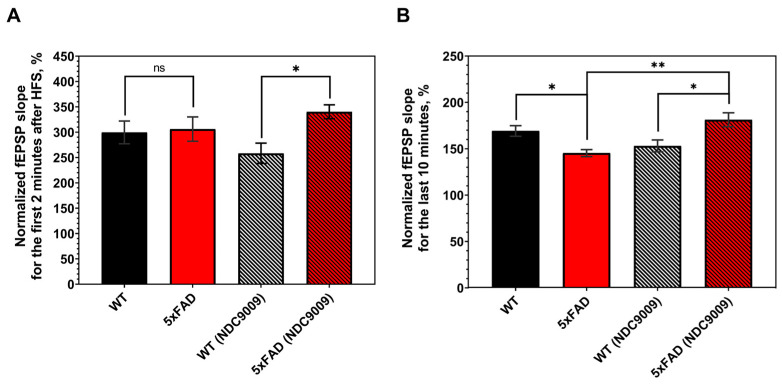
Positive allosteric modulators of SERCA pump (NDC-9009) restores LTP impairment in 6-month-old 5xFAD mice. (**A**) Normalized fEPSP slope (%) for post-tetanic potentiation (2 min after high-frequency stimulation). (**B**) Normalized fEPSP slope (%) for long-term potentiation (last 10 min after high-frequency stimulation). Groups: WT: *n* = 10; 5xFAD: *n* = 8; WT (NDC-9009): *n* = 7; 5xFAD (NDC-9009): *n* = 9. Statistical analysis was performed using a Kruskal–Wallis test with a post hoc Dunn’s test for PTP analysis and one-way ANOVA with a post hoc Tukey’s test for LTP analysis (*: *p* < 0.05, **: *p* < 0.01). Data are presented as the mean ± SEM.

**Table 1 ijms-24-13973-t001:** Biochemical properties of SERCA PAMs.

Compound	MW	Structure	% SERCA Activation @ 2 µM	Solubility (µM) in PBS @ pH 7.4
NDC-9009	305	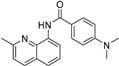	17.1 ± 1.0	0.23
NDC-9033	320	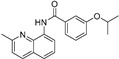	16.5 ± 2.5	0.05
NDC-9136	320	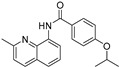	13.9 ± 2.1	0.10
NDC-9342	426	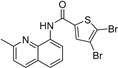	28.4 ± 1.2	0.03

**Table 2 ijms-24-13973-t002:** Results of evaluation of the neuroprotective effect of SERCA PAMs. Data are expressed as the mean ± SEM. Statistical significance was determined by using a Conover–Iman test in the analysis of the spines’ head area and Dunn’s test for the calcium signal and curve slope (* *p* < 0.05, ***p* < 0.001, *** *p* < 0.001 compared to control).

Criterion	Control	NDC-9009	NDC-9033	NDC-9136	NDC-9342
Spines’ head area in normal conditions, µm^2^	0.583 ± 0.025	0.588 ± 0.026	0.511 ± 0.026 *	0.598 ± 0.029	0.430 ± 0.019 ***
Spines’ head area in conditions of amyloid toxicity, µm^2^	0.481 ± 0.019	0.576 ± 0.041 *	0.530 ± 0.020	0.560 ± 0.028 *	0.563 ± 0.031 *
Calcium signal (dF/F0)	7.2 ± 0.6	4.78 ± 0.3 ***	7.97 ± 0.7	4.9 ± 0.5 **	5.9 ± 0.4 *
Curve slope (tanα)	0.061 ± 0.006	0.021 ± 0.002 ***	0.048 ± 0.006	0.016 ± 0.002 ***	0.026 ± 0.003 ***

## Data Availability

Not applicable.

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
