# Peer review of "Positive Allosteric Modulators of SERCA Pump Restore Dendritic Spines and Rescue Long-Term Potentiation Defects in Alzheimer’s Disease Mouse Model"

_ijms, 2023, doi:10.3390/ijms241813973_

Round 1

Reviewer 1 Report

This manuscript describes an interesting set of experiments designed to test the neuroprotective benefits of a novel class of SERCA pump modulators as candidate therapeutic compounds for treating AD. Based on promising recent studies by this group and others that SERCA modulators can protect neurons against calcium overload associated with amyloid toxicity, the investigators have now analyzed a set of newly identified SERCA positive allosteric regulators (PAMs) for their ability to protect against some of the deleterious effects caused by exogenous amyloid beta in primary hippocampal neurons and in the 5XFAD mouse model. This combination of in vitro and in vivo strategies is particularly noteworthy, as it also demonstrates that the novel PAMs can be safely administered to intact animals for prolonged periods.  The fact that one of the PAMs (NDC-9099) both protected against the loss of dendritic spine morphology in culture neurons and mitigated the loss of normal LTP responses in brain slice preparations supports the conclusion that this compound (or others like it) may represent a promising therapeutic strategy for preventing or treating AD, particularly at earlier stages of the disease.

The manuscript is well written (no issues noted); the figures are clear and describe appropriately in the text, and the conclusions are reasonable.

There are only two related points that the authors should address in somewhat more detail (e.g. in their discussion.

As the authors note (referring to figures 3 & 4), LTP measurements showed that treating 5XFAD mice with NAD09009 caused a substantial increase in values, while treating Wt mice with the compound caused a moderate reduction. This puzzling contrast deserves some discussion; what aspect of the progressive amyloid pathology seen in 5XFAD mice might render them more susceptible to the effects of SERCA PAMs?

Author Response

We would like to thank the referee for constructive comments that helped us to improve the paper.  Please find our response below:

  1. As the authors note (referring to figures 3 & 4), LTP measurements showed that treating 5XFAD mice with NAD09009 caused a substantial increase in values, while treating Wt mice with the compound caused a moderate reduction. This puzzling contrast deserves some discussion;what aspect of the progressive amyloid pathology seen in 5XFAD mice might render them more susceptible to the effects of SERCA PAMs?

Answer: In the presented graphs, we observe a trend indicating a reduction in PTP and LTP levels in WT mice following NDC-9009 administration; however, this trend does not reach a statistically significant level. Consequently, we have opted to omit these sentences: “Concurrently, the WT (NDC-9009) group displayed a propensity for a diminished PTP value in contrast to the wild type mouse control group” and “In contrast, wild-type mice in the experimental group exhibited a tendency to reduce the value of LTP in comparison to the control group”, to avoid misunderstanding. As an answer to your question, we can provide the following explanation: SERCA PAMs are compounds designed to enhance the activity of SERCA pumps, thereby influencing calcium regulation within neurons. In the context of 5XFAD mice or Alzheimer's disease models, the accumulation of amyloid-beta plaques may lead to elevated concentration of calcium [Role of calcium in the pathogenesis of Alzheimer’s disease and transgenic models] and contribute to neuronal dysfunction. SERCA PAMs may be effective in such models because they can potentially restore calcium homeostasis, which is critical for proper neuronal function. Therefore, one hypothesis could be that the disrupted calcium homeostasis in 5XFAD mice makes them more susceptible to the beneficial effects of SERCA PAMs, as these compounds could potentially mitigate the calcium dysregulation and associated neuronal dysfunction caused by amyloid pathology. It is also worth noting that in wild-type mice, the effect of SERCA positive allosteric modulators may have a negative effect. A possible explanation for this could be the different levels of intracellular calcium in WT and 5xFAD. In the case of 5xFAD, significant increases in calcium concentration lead to SERCA activations with PAMs and induce a decrease in intracellular calcium levels, which normalize disturbed neuronal function. In WT mice positive allosteric modulators (PAMs) are assumed not to be active, and therefore calcium levels don't decrease above normal level. Hence, PAMs do not exert an impact on basal activity in WT mice, consequently yielding no significant differences in PTP and LTP values between the control and experimental groups in WT mice. However, it's important to note that this is a hypothesis, and further research would be needed to confirm this mechanism and its relevance to the specific context of 5XFAD mice and SERCA PAMs. For this reason, we have not added this text to the discussion section.

Reviewer 2 Report

The methods section requires more information.  Were both sexes of animals used?  Did the pups used for in vitro cultures have their sexes determined?  Why were newborn mice used if this examined Alzheimer's Disease?  Can this be called an Alzheimer's model with animals of such a young age?  Why was this particular area of the hippocampus chosen?  What bregma/x-y-z coordinates were used? 

Additionally, several of the figures have text that is too small to be legible.

Lastly, the discussion needs more discussion of the results currently being presented, in lieu of a repetition of what is stated in the introduction.

There are multiple small issues throughout the paper with regards to the English language.  Editing is needed for the following:  misspellings, ex. 'Neuronsl' in the abstract, run on sentences, inclusion of articles, ex. 'recently the FDA' as opposed to 'Recently FDA', defining all acronyms, unnecessary commas, unnecessary hyphens, consistency in measurement units, and unnecessary spaces.

Author Response

Thank you very much for your comments. Below, you will find a point-by-point response:

1.The methods section requires more information.  Were both sexes of animals used?  Did the pups used for in vitro cultures have their sexes determined?  Why were newborn mice used if this examined Alzheimer's Disease?  Can this be called an Alzheimer's model with animals of such a young age?  Why was this particular area of the hippocampus chosen?  What bregma/x-y-z coordinates were used?

Answer: Primary hippocampal neuronal cultures were prepared as previously described (Popugaeva E., 2015). The entire hippocampus from postnatal day 0–2 FVB mice of both sexes was used for primary neuronal culture. To simulate Alzheimer's disease, neuronal cultures were incubated for 72 hours with oligomeric Aβ42 fractions at low concentrations, following the previously published protocol (Popugaeva E., 2015). This model was also used in other papers to model Alzheimer's disease in vitro (Zhang H., 2016, Pchitskaya E., 2022 and etc). For ex vivo part of work with Long-Term Potentiation (LTP) experiments were used male 5xFAD mice at 6 months of age. Necessary corrections to be more clear about used models introduced to the publication text.

  1. Additionally, several of the figures have text that is too small to be legible.

Answer: Thank you for bringing this issue to our attention. We have resized the text on Figures 1 and 2 to ensure that it is sufficiently readable in the revised version of the manuscript.

  1. Lastly, the discussion needs more discussion of the results currently being presented, in lieu of a repetition of what is stated in the introduction.

Answer: We have implemented the requisite modifications to the discussion section of the manuscript. Added more detailed information in the discussion about another target SERCA PAMs: “Also, a study conducted on a pyridone derivative was found to activate the SERCA2a isoform. Furthermore, it was observed that the pyridone derivative stimulated the Ca2+-dependent ATPase activity of cardiac SR (sarcoplasmic reticulum) vesicles. This indicates that the derivative enhances the ability of SR to transport calcium ions, which is an essential process for proper cardiac function..”. Additionally, added more information about results: “We demonstrated the in vitro calcium ATPase activity using SERCA PAMs on the HEK293T cell line. Additionally, we evaluated the neuroprotective potential of SERCA PAMs…”. Finally, added information about limitations of the study: “Several limitations of this study should be noted…”.

  1. Comments on the Quality of English Language. There are multiple small issues throughout the paper with regards to the English language. Editing is needed for the following:  misspellings, ex. 'Neuronsl' in the abstract, run on sentences, inclusion of articles, ex. 'recently the FDA' as opposed to 'Recently FDA', defining all acronyms, unnecessary commas, unnecessary hyphens, consistency in measurement units, and unnecessary spaces.

Answer: Thank you for your comment. We have implemented the necessary improvements to the text.

Reviewer 3 Report

The manuscript "Positive allosteric modulators of SERCA pump restore dendritic spines and rescue long-term potentiation defects in Alzheimer`s disease mice model"" is interesting and well written. The results are not misleading. 

Introduction:
- Please provide a clear hypothesis in the introduction section

- I would suggest to mention in the introduction the immune system, it interacts a lot with synapses it worth mentioning

You mention memantine as a good first-line medication. It would be interesting to develop this point because I have seen many controversies about it. Some scientists compare memantine to a sugar pill in terms of its effectiveness in Alzheimer's.

Resutls:
2.3: You have incubated cells with DMSO, did you validate the maximal dose of DMSO tolerated by the primary cell culture? 

Table 2: you used SD, in other results you used SEM, is there a reason ?   

Figure 4: You used 6-month-old 5xFAD, why did you choose this model over APP/PS1 mentioned in the introduction. 5xFAD is a quite controversial model especially in behavioral tests and disease onset. Please provide a photo showing amyloid at 6 month for these mice. I can seem trivial, but as the model is not as reliable as APP/PS1, it will prove that you have amyloid burden at 6-month-old. 

Materials and Methods: 
Could you describe in this section or in another how did you choose the dose, how did you validate it? 

Mice were treated during the "pre-symptomatic" phase and examined at 6 month, did you try dosing the mice around 6-7 month with the same protocol to see if the molecule can reverse the symptoms?

Discussion:

Please develop the discussion, could you discuss the limitations of this study and the perspectives? what is the next step, what validations are necessary for the molecules? what study need to be done in which model,.....

Overall the article is well written, and sounds scientific. I have minor comments, but they need to be addressed. 

I would also suggest to improve the presentation of the figures and if it's possible provide a graphical abstract.

Thanks 

Author Response

Thank you very much for detailed and constructive comments.  We addressed most of them and we hope that you will find the revised paper satisfactory. Point-by-point response to your comments  is below:

Introduction:
- Please provide a clear hypothesis in the introduction section

Answer: We have implemented the necessary improvements to the introduction section of the manuscript. Adjustments have been made: “In this study, we aim to study the activity of a new class of SERCA PAMs in vitro and their potential protective, and therapeutic effects on in vitro and ex vivo AD models”.

- I would suggest to mention in the introduction the immune system, it interacts a lot with synapses it worth mentioning.

Answer: Adjustments have been made in introduction: “However, a substantial unmet need exists for the development of safer therapies for Alzheimer's disease (AD) that can potentially offer clinical benefits to patients. It is noteworthy that the immune system interacts with synapses, with immune cells and mediators participating in synapse elimination during development and contributing to synaptic plasticity during adulthood (Desai B.N., 2014; Sebastião A.M., 2015). Therefore, in the context of drug development, it is imperative to consider their impact on the immune system.”.

-You mention memantine as a good first-line medication. It would be interesting to develop this point because I have seen many controversies about it. Some scientists compare memantine to a sugar pill in terms of its effectiveness in Alzheimer's.

Answer: Added more detailed information in the introduction: “Memantine is a noncompetitive NMDA glutamate receptor antagonist. Memantine is also a dopamine agonist and increases dopaminergic transmission. In cognitive terms, me-mantine primarily increases attention and episodic memory (when attention improves, general memory improves).Memantine offers no benefit in mild AD, but it improves symptoms in patients with moderate to severe AD and exhibits a favorable safety and tol-erability profile (McShane R., 2019)”.

Results:

2.3: You have incubated cells with DMSO, did you validate the maximal dose of DMSO tolerated by the primary cell culture?

Answer: DMSO was used as the initial solvent for the Aβ synthetic peptides and NDC compounds. Then, it was diluted extensively to reach the working concentration. For  Aβ the final concentration was only 0.1 % in the control group, and it is noteworthy that this concentration has been reported to have no impact on neurons, as documented in the study by Zhang, Chen et al. (2017).

Table 2: you used SD, in other results you used SEM, is there a reason ?

Answer: We sincerely apologize for the typographical error in the table, where 'SD' was mistakenly used instead of 'SEM.' We have subsequently made the requisite corrections to the article's text.

Figure 4: You  , why did you choose this model over APP/PS1 mentioned in the introduction. 5xFAD is a quite controversial model especially in behavioral tests and disease onset.

Answer: The APP/PS1 model, used in the study (Dahl et al., 2023), does not exhibit the hallmark characteristic of abeta-amyloid accumulation associated with Alzheimer's disease. Consequently, the 5xFAD mouse line was selected for this study.

Please provide a photo showing amyloid at 6 month for these mice. I can seem trivial, but as the model is not as reliable as APP/PS1, it will prove that you have amyloid burden at 6-month-old.

Answer:   Amyloid images for wild and 5xFAD mice at 6 months of age are now included as Supplementary Fig 1.   And added this text to manuscript: “We have conducted experiments to assess post-tetanic (PTP) and long-term potentiation (LTP) in control and experimental groups of wild-type (WT) and Alzheimer's mice (5xFAD), characterized by the production and aggregation of beta-amyloid protein (Figure 3; representative images of amyloid plaques in the brains of 6-month-old 5xFAD and WT mice are presented in Figure S1).”

Materials and Methods:

Could you describe in this section or in another how did you choose the dose, how did you validate it?

Answer:  The appropriate concentrations of NDC compounds were established in prior studies (Krajnak & Dahl, 2018; Dahl et al., 2023), and these concentrations were employed in our current investigation. References to these relevant previous studies have been incorporated into the "Materials and Methods" section of the manuscript.

Mice were treated during the "pre-symptomatic" phase and examined at 6 months, did you try dosing the mice around 6-7 month with the same protocol to see if the molecule can reverse the symptoms?

Answer: We have not conducted experiments with mice in the 6-7 month age range using the same protocol to assess the molecule's effects on symptom reversal. Our studies concerning impaired long-term potentiation have been concentrated solely on mice aged six months. However, we are considering future studies in that direction.

Discussion:

Please develop the discussion, could you discuss the limitations of this study and the perspectives? what is the next step, what validations are necessary for the molecules? what study need to be done in which model,.....

Answer: We have implemented the requisite modifications to the discussion section of the manuscript. Kindly refer to the final version of the revised manuscript. Added more detailed information in the discussion about another target SERCA PAMs: “Also, a study conducted on a pyridone derivative was found to activate the SERCA2a isoform. Furthermore, it was observed that the pyridone derivative stimulated the Ca2+-dependent ATPase activity of cardiac SR (sarcoplasmic reticulum) vesicles. This indicates that the derivative enhances the ability of SR to transport calcium ions, which is an essential process for proper cardiac function..”. Additionally. added more information about results: “We demonstrated the in vitro calcium ATPase activity using SERCA PAMs on the HEK293T cell line. Additionally, we evaluated the neuroprotective potential of SERCA PAMs…”. Finally, added information about limitations of the study: “Several limitations of this study should be noted…”.

Overall the article is well written, and sounds scientific. I have minor comments, but they need to be addressed. I would also suggest to improve the presentation of the figures and if it's possible provide a graphical abstract.

Answer: Thank you for your feedback on the manuscript. We appreciate your positive assessment of its overall quality. We have implemented the necessary revisions to improve the manuscript.